# Bacteriophage-antibiotic combination therapy against extensively drug-resistant *Pseudomonas aeruginosa* infection to allow liver transplantation in a toddler

Post-operative bacterial infections are a leading cause of mortality and morbidity after ongoing liver transplantation. Bacteria causing these infections in the hospital setting can exhibit high degrees of resistance to multiple types of antibiotics, which leads to major therapeutic hurdles. Alternate ways of treating these antibiotic-resistant infections are thus urgently needed. Phage therapy is one of them and consists in using selected bacteriophage viruses – viruses who specifically prey on bacteria, naturally found in various environmental samples – as bactericidal agents in replacement or in combination with antibiotics. The use of phage therapy raises various research questions to further characterize what determines therapeutic success or failure. In this work, we report the story of a toddler who suffered from extensively drug-resistant *Pseudomonas aeruginosa* sepsis after liver transplantation. He was treated by a bacteriophage-antibiotic intravenous combination therapy for 86 days. This salvage therapy was well tolerated, without antibody-mediated phage neutralization. It was associated with objective clinical and microbiological improvement, eventually allowing for liver retransplantation and complete resolution of all infections. Clear in vitro phage-antibiotic synergies were observed. The occurrence of bacterial phage resistance did not result in therapeutic failure, possibly due to phage-induced virulence tradeoffs, which we investigated in different experimental models.

Post-operative bacterial infections are a leading cause of mortality and morbidity after ongoing liver transplantation[1]. The spread and selection of antimicrobial resistance mechanisms among bacteria over the last decades now lay a considerable burden on global health, being directly responsible for more than a million yearly deaths[2]. These resistant bacteria have also become more frequently involved in the aforementioned post-operative infections[1]. *Pseudomonas aeruginosa* is a bacterial species responsible for a specifically high number of hospital-acquired bacterial infections, in part because of its ability to thrive in microbial "reservoirs" in the hospital setting[3]. *P. aeruginosa*

can be considered a critical clinical pathogen for its numerous virulence factors, its ability to persist in a self-protective biofilm state, and for both its intrinsic and acquired mechanisms of multi-drug resistance[4]. Phage therapy, the therapeutic use of lytic (bacterio)phages for their bactericidal properties, has regained interest as a complementary tool in the fight against multi-drug-resistant bacterial infections[5], including those due to *P. aeruginosa*[6]. Phages used in phage therapy are viruses that specifically infect and lyse a given bacterial species, or a limited number of related bacterial species. Phage therapy's efficacy and safety in the treatment of human

e-mail: brieuc.vannieuwenhuyse@uclouvain.be

infections through various administration routes are globally promising and have been the subject of a growing interest in recent clinical research[7].

In September 2018, a male toddler suffering from biliary atresia was transferred from Algeria to the Saint-Luc University Hospital (*Cliniques universitaires Saint-Luc - CUSL*). ABO incompatible (ABOi) living-donor liver transplantation (LDLT, day 0) was performed using an adapted protocol, including rituximab and plasmapheresis for optimal immunosuppression (Fig. 1).

On day 20 post-LDLT, routine active surveillance (rectal swab) showed nosocomial acquisition of *Pseudomonas aeruginosa*. The isolated *P. aeruginosa* strain's antibiogram suggested an extensively drug-resistant (XDR) phenotype, with susceptibility to colistin only and intermediate susceptibility to aztreonam (data not shown). On day 53, the toddler presented intrahepatic biliary strictures, multiple hepatic abscesses, cholangitis, and severe septicemia caused by XDR

*P. aeruginosa* (Fig. 1). Despite intravenous (IV) antibiotic therapy with gentamycin, colistin and high doses of aztreonam, blood and abscess samples continued to grow XDR *P. aeruginosa*. Concomitantly, the C-reactive protein (CRP) ranged from 89 to 189 mg/L. The child remained clinically septic and unstable and, for the fifth time since LDLT, had to be admitted to the PICU (Fig. 1).

The ongoing systemic infection was a contraindication for retransplantation. Because the antibiotic therapy failed and the child's situation was critical, the decision was taken at day 58 post-LDLT to drain the hepatic abscesses, and to initiate phage therapy (PT) under the umbrella of article 37 (unmet medical need) of the Declaration of Helsinki and after Ethics Committee approval (Fig. 1). BFC1 ("bacteriofaagcocktail 1") is a bacteriophage (phage) cocktail produced by the Queen Astrid Military Hospital (QAMH) in Brussels, containing one *Staphylococcus aureus* phage (ISP) and two *P. aeruginosa* phages (PNM and 14-1)[8]. Before clinical application, the phage preparation was

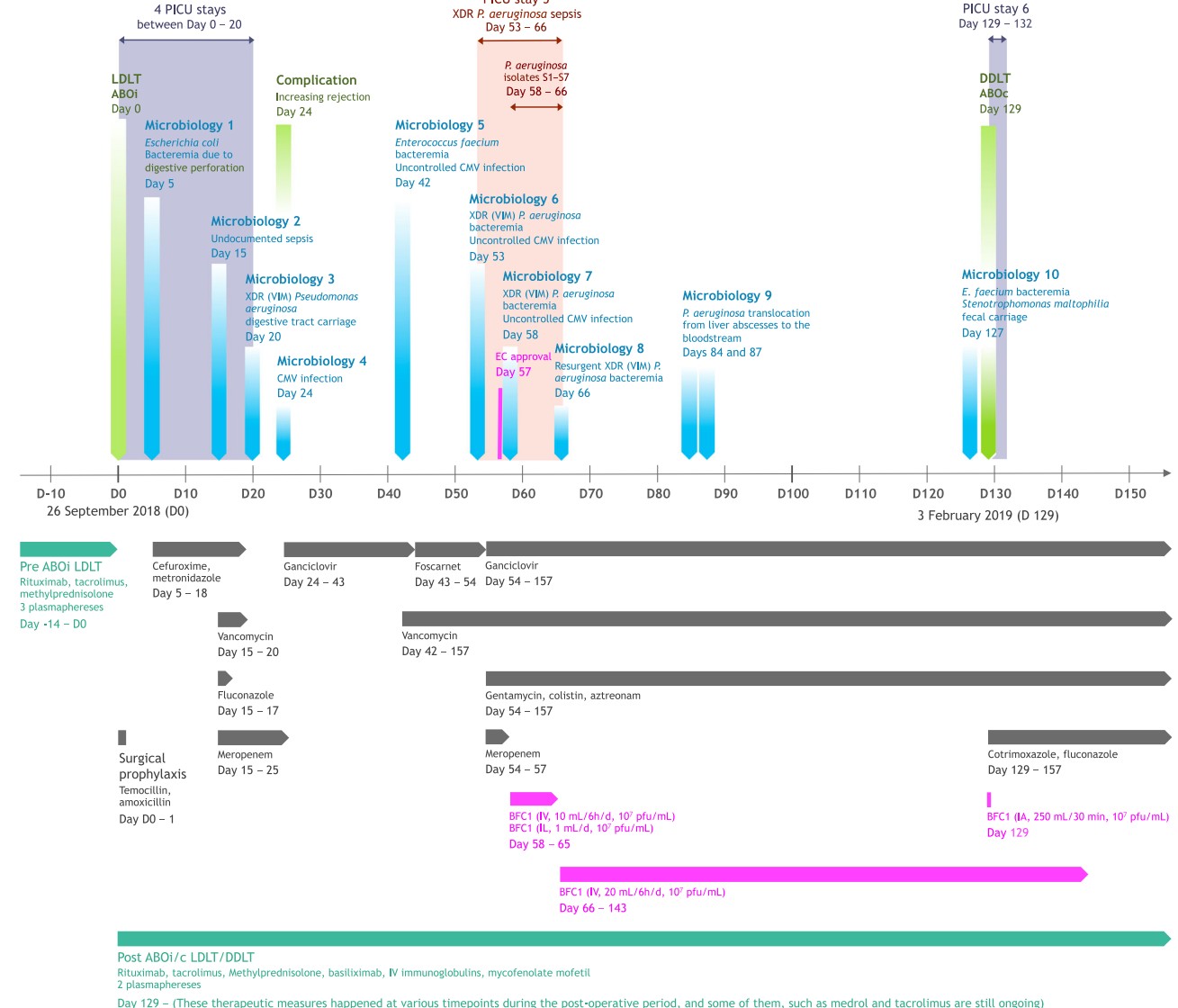

**Fig. 1 | Clinical timeline.** Timeline of the most relevant surgical procedures (light green), microbiological results (light blue), antibiotic therapies (dark gray), pre- and post-liver transplantation protocols (blue-green), and phage therapy (magenta). The child developed several early-onset post-operative complications, including liver rejection, anaphylactic shock on rituximab, biliary digestive anastomosis perforation followed by an *Escherichia coli* and *Enterococcus faecium* sepsis, and cytomegalovirus (CMV) infection. At day 53 post-LDLT, he entered a severe septicemia due to XDR *Pseudomonas aeruginosa*. ABOc ABO compatible, ABOi ABO incompatible, BFC1 "bacteriofaagcocktail 1", CMV cytomegalovirus, DDLT deceased-donor liver transplantation, EC ethics committee, IA intra-abdominal, IL intralesional, IV intravenous, LDLT living-donor liver transplantation, pfu plaque forming unit, PICU pediatric intensive care unit, VIM Verona integron-encoded metallo-β-lactamase, XDR extensively drug-resistant.

analyzed and approved by the Belgian Institute for Public Health (Sciensano), according to the quality criteria set in the general monograph (v1.0) for the production of phage pharmaceutical ingredients[9]. All three phages composing BFC1 have been sequenced and fully characterized. None of these phages contain any known genetic determinants coding for lysogeny, toxins, or antibiotic resistance. All three phages are considered strictly virulent. Based on the genome analysis, Sciensano issued a "green" genomic passport, certifying the absence of any known genetic determinants that would make these phages unsuitable for therapeutic use. BFC1 was purified using EndoTrap® affinity matrix and then certified by Sciensano to contain <5 EU/mL of endotoxins.

Ten milliliters (1 mL/kg) of BFC1 [$10^7$ plaque-forming units (pfu)/mL] were administered by a daily 6-hour IV infusion. Daily intralesional (IL) injections of 1 mL BFC1 were performed through the patient's biliary drainage catheter, but were suspended after 7 days, because they caused abdominal discomfort in the child. Thirty-six hours after PT initiation, two subsequent blood cultures were negative. However, between days 4 and 8 after initiation of PT, blood and abscess pus cultures revealed the presence of *P. aeruginosa*. At day 8 (day 66 post-LDLT), the IV phage dose was doubled (2 mL/kg), and this for the remaining duration of the treatment. This led to the durable eradication of *P. aeruginosa* from the bloodstream; although two transient *P. aeruginosa* translocations from liver abscesses to the bloodstream were observed on day 27 and day 30 after PT initiation, they both disappeared in less than 24 h without new therapeutic measures, and had no impact on the child's stable clinical state.

PT combined with antibiotics controlled the *P. aeruginosa* bloodstream infection, but did not eradicate *P. aeruginosa* in the hepatic lesions. The child remained clinically stable outside of the PICU with intermittent fever and persistent inflammation. This stable state was maintained until a postmortem ABO compatible (ABOc) liver was available for retransplantation, after 72 days of IV PT. Two days prior to retransplantation, the child developed a bacteremia with *Enterococcus faecium*, which was successfully treated with vancomycin. *Stenotrophomonas maltophilia* fecal carriage was also newly identified.

During the deceased-donor liver transplantation (DDLT), 250 mL of BFC1 was used to rinse the abdominal cavity (Fig. 1 and Supplementary Fig. 1). This procedure was well tolerated and IV PT was continued for two more weeks. Not surprisingly, culture of tissue samples from the removed diseased liver revealed the presence of *P. aeruginosa*, *E. faecium*, and *S. maltophilia*. The child was also treated with a wide combination of antibiotics (cotrimoxazole, vancomycin, colistin, ganciclovir, fluconazole, aztreonam, and gentamycin) for 28 days post-DDLT (Fig. 1). In total, the child received 86 uninterrupted days of IV PT (72 days pre- and 14 days post-DDLT) (Fig. 1). No adverse event related to the IV phage therapy was observed during hospitalization.

More than two years after this second liver transplantation, the child is doing fine on immunosuppressive medication, with normal liver function, no transplant rejection, and no further infectious outbreaks. Total clearance of *P. aeruginosa* colonization was observed.

In this work, we investigate the net contribution of PT in the child's favorable outcome through three phenomena that shed light on the in vivo antibacterial action of phages, in view of future salvage therapies and clinical trials: (i) in vivo bacterial phage resistance (BPR) emergence and possible incurred phage-induced virulence tradeoffs (PIVT), (ii) phage-antibiotic synergy (PAS), and (iii) phage immune neutralization (PIN).

## Results and discussion

Seven XDR *P. aeruginosa* isolates, retrieved from four blood cultures (Pa1$_{BS}$, Pa2$_{BR}$, Pa4$_{BR}$, and Pa7$_{BS}$) and three hepatic abscess pus cultures (Pa3$_{LS}$, Pa5$_{LR}$, and Pa6$_{LR}$) were retained for further analysis (Table 1). These isolates were sampled during what was arguably the most critical period in the history of this case, from day 58 to day 66 post-LDLT.

Only the first one (Pa1$_{BS}$) was obtained before (4 h) PT initiation, whereas the other six (Pa2$_{BR}$-Pa7$_{BS}$) were retrieved during the first week of PT. Due to the sense of urgency brought on by XDR *P. aeruginosa* bacteremia, the susceptibility of the *P. aeruginosa* isolates to the phages of the BFC1 cocktail could not be assessed prior to the start of PT. Two days later, we observed that only one of the three phages that make up cocktail BFC1, *P. aeruginosa* phage PNM, showed confluent lytic activity on pre-PT *P. aeruginosa* blood isolate Pa1$_{BS}$ and subsequent isolates Pa3$_{LS}$ and Pa7$_{BS}$. Four of these seven isolates were phenotypically resistant to all phages present in BFC1: Pa2$_{BR}$, Pa4$_{BR}$, Pa5$_{LR}$, and Pa6$_{LR}$ (Table 1).

**Antimicrobial resistance, phage resistance, induced tradeoffs**

Whole genome sequencing (WGS) of the seven *P. aeruginosa* isolates considered was performed by combining long-read and short-read sequencing technologies[10]. WGS and subsequent single-nucleotide polymorphism (SNP) analysis confirmed that isolates Pa1$_{BS}$–Pa7$_{BS}$ all belong to the same *P. aeruginosa* strain with over 99.9999% ANIb similarity and a maximum distance of three SNPs (Supplementary Data 1). In silico analysis revealed that the strain belongs to sequence type ST111 (serotype O12), known for its prevalence in nosocomial *P. aeruginosa* infections and its extensively drug-resistant profile[11]. Five of the isolates harbor the *bla*$_{VIM-2}$ gene coding for a VIM (Verona Integron-encoded Metallo-beta-lactamase) type carbapenemase, which is a common feature of this high-risk clone[11]. WGS revealed that each isolate contained six to eight genetic determinants known to confer antimicrobial resistance (Supplementary Fig. 2). The distribution of these genetic determinants among the isolates does not indicate that any of them would have been selected by the administration of antibiotics to the patient. The circular chromosomic views (CCV) of the bacterial genomes are depicted in Fig. 2. Additionally, WGS analysis showed that the four strains expressing BPR possess genetic alterations in the 5.8–5.9 Megabases (Mb) chromosomic region, which affect the development of the Type IV pili (T4P) complex (Supplementary Data 1 and 2). This region contains various T4P-related genes such as *pilM*, *pilO2*, *pilP*, and *pilV*[12]. In isolate Pa6$_{LR}$, T4P variation consisted of an SNP in the *pilB* locus, whereas in the three other phage-resistant isolates (Pa2$_{BR}$, Pa4$_{BR}$, and Pa5$_{LR}$), it consisted in a regional deletion in a putative *P. aeruginosa* genomic island (PAGI). In addition, alternative or complementary ways of achieving T4P-modification can be suspected, for example, *pilM* deactivation by IS5 transposase activity in isolate Pa2$_{BR}$, or a PAGI-90 deletion in isolate Pa5$_{LR}$. Phage PNM was shown to require the presence T4P for successful infection[13]. Acquiring BPR through genetic alteration of outer phage adsorption determinants is one of the most frequently described mechanisms of BPR acquisition[14,15]. It is thus possible that a component of the T4P structure serves as a specific adsorption site for phage PNM.

The retrieval of blood-circulating *P. aeruginosa* isolates, which show in vitro insensitivity to all administered antibiotics and phages that were administered (Pa2$_{BR}$ and Pa4$_{BR}$), could have been considered as a sign of impending therapeutic failure. Yet, PT initiation led to the immediate and sustained improvement of the child's clinical state, and eventually to the permanent exclusion of *P. aeruginosa* from the bloodstream. It has often been suggested that besides their primary lytic effect, phages can also indirectly threaten bacterial prosperity by exerting a deteriorating selective pressure. BPR can come with a fitness cost, as it can require genetic variations which can lead to the loss of key survival elements, like intrinsic virulence or antimicrobial resistance[16,17]. Facing such fitness costs, bacteria are "caught in a vice" where escaping phage lysis results in recovering previous antimicrobial susceptibility and/or losing intrinsic virulence. Since T4P regulates virulence in *P. aeruginosa*[18], the occurrence of PIVT was plausible. All seven isolates were analyzed using an OmniLog® system in identical conditions, which did not highlight any significant difference in bacterial growth rate between the strains over 72 h

**Table 1 | Phage susceptibility of seven *Pseudomonas aeruginosa* isolates retrieved before (Pa1BS) and during (Pa2BR–Pa7BS) phage therapy**

| Name | Isolation site | Isolation date | Time relative to PT start | Phage lytic activity | | |
|------|----------------|----------------|----------------------------|-------------------------|-------------------------|---------|
| | | | | PNM (10$^9$ pfu/mL) | PNM (10$^7$ pfu/mL) | PNM EOP |
| Pa1$_{BS}$ | Blood | 23 Nov 18 | −4 h | ++++ | ++ | 0.3 |
| Pa2$_{BR}$ | Blood | 24 Nov 18 | +8 h | − | − | − |
| Pa3$_{LS}$ | Liver | 24 Nov 18 | +11 h | ++++ | +++ | 0.2 |
| Pa4$_{BR}$ | Blood | 24 Nov 18 | +4 days | − | − | − |
| Pa5$_{LR}$ | Liver | 27 Nov 18 | +4 days | − | − | − |
| Pa6$_{LR}$ | Liver | 27 Nov 18 | +4 days | − | − | − |
| Pa7$_{BS}$ | Blood | 1 Dec 18 | +8 days | ++++ | +++ | 0.27 |

The nomenclature of these isolates consists of: [Pa] for *P. aeruginosa*; [number] for chronological order of isolation; [first letter: B/L] for origin of isolate, either from Blood culture or Liver abscess pus culture; [second letter: S/R] for phenotypic Susceptibility or Resistance of the isolate to phage PNM. *P. aeruginosa* isolates Pa1BS, Pa2BR, Pa4BR, and Pa7BS were retrieved from blood samples harvested on days 57, 58, 61, and 65 post-LDLT, respectively. *P. aeruginosa* isolates Pa3LS, Pa5LR, and Pa6LR were isolated from liver abscess pus on days 58 (Pa3LS) and 61 (Pa5LR and Pa6LR). Phage susceptibility was determined by spot test.
*EOP* efficiency of plating, *pfu* plaque forming unit, ++++ confluent lysis, +++ semi-confluent lysis, ++ opaque lysis, − no lysis

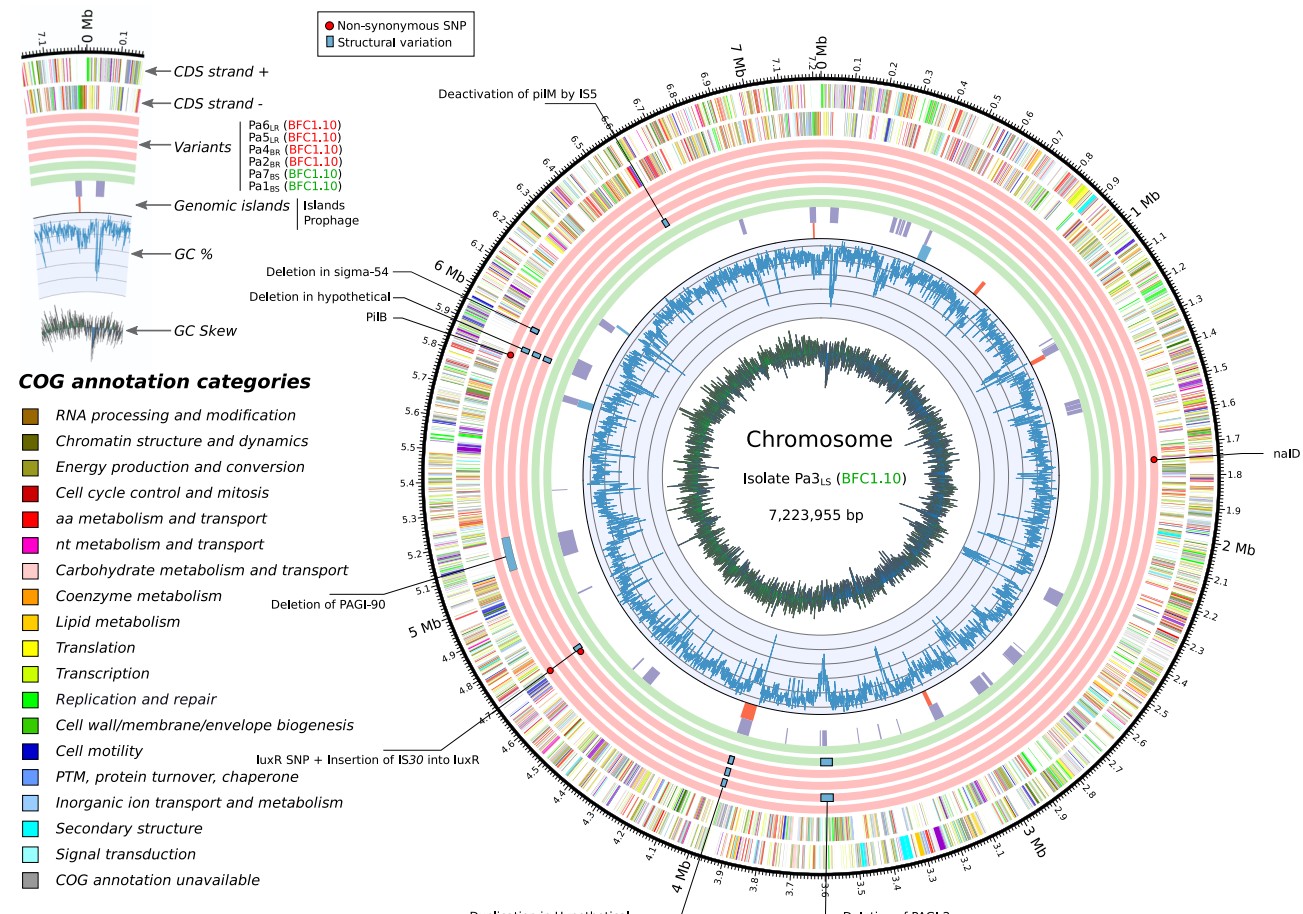

**Fig. 2 | Circular chromosomic view (CCV) of the bacterial genomes of seven *Pseudomonas aeruginosa* isolates retrieved just before (Pa1$_{BS}$) and during (Pa2$_{BR}$–Pa7$_{BS}$) phage therapy.** The CCV algorithm chose strain Pa3$_{LS}$ as reference (inner circle). From the inner to outer rings of the CCV, two green rings display the genomic variations in phage PNM-susceptible isolates Pa1$_{BS}$ and Pa7$_{BS}$, and four outer red rings display the genomic variations in phage PNM-resistant isolates Pa2$_{BR}$, Pa4$_{BR}$, Pa5$_{LR}$, and Pa6$_{LR}$. The two multi-colored outer rings display the protein annotations (categories) as present in the database of Clusters of Orthologous Groups of proteins (COGs). aa amino acid, bp basepairs, CDS coding sequence, IS insertion sequence, Mb megabases, nt nucleotide, PAGI *Pseudomonas aeruginosa* genomic island, PTM post-translational modification; SNP single-nucleotide polymorphism.

(Supplementary Fig. 3). Therefore, we conducted *Galleria mellonella* (*Gm*) virulence assays (Fig. 3a–c). Batches of *Gm* larvae were inoculated with any of the seven retained *P. aeruginosa* isolates (or with a phosphate-buffered saline (PBS) control), and their viability (activity

score) was monitored hourly to look for possible differences in virulence between the different isolates (Supplementary Fig. 4). In addition, a second independent assay was performed to assess the virulence of the isolates. We performed a cell viability assay using HeLa

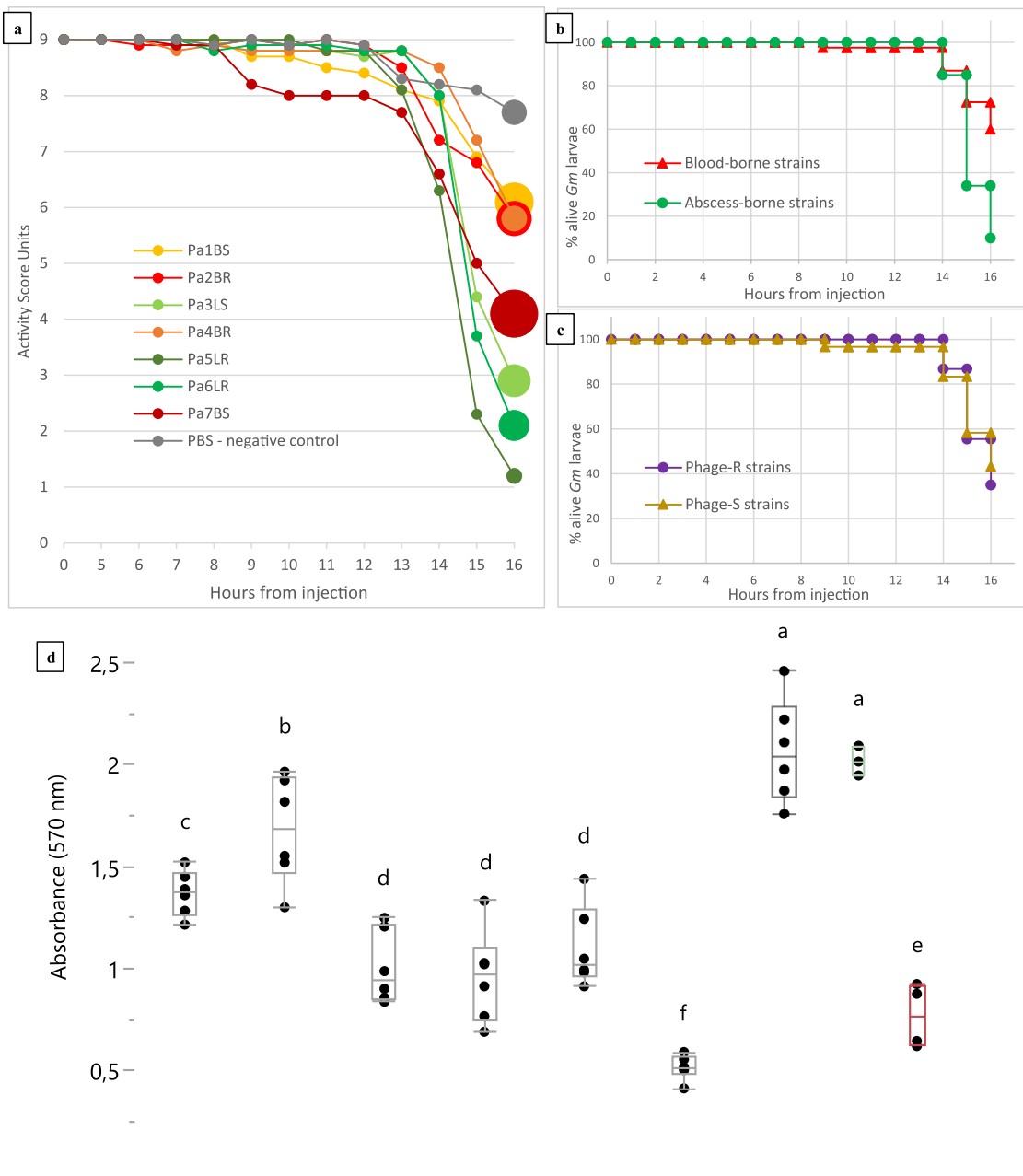

**Fig. 3 | Virulence assays. a** Chronological evolution of the mean activity score of *Gm* larvae injected with phosphate-buffered saline (control) or any of the seven *Pseudomonas aeruginosa* isolates retrieved just before (Pa1$_{BS}$) and during (Pa2$_{BR}$–Pa7$_{BS}$) phage therapy. Full lines represent the mean activity scores of *Gm* larvae injected with any of four PNM-resistant *P. aeruginosa* isolates (Pa2$_{BR}$ and Pa4$_{BR}$–Pa6$_{LR}$) or phosphate-buffered saline (PBS). Dashed lines were used for the three PNM-susceptible isolates (Pa1$_{BS}$, Pa3$_{LS}$, and Pa7$_{BS}$). Note that first shown measurement after injection starts at +5 h. Real scale standard deviations were not visually implementable, but the dot width of each final mean measurement correlates with its standard deviation. $n = 10$ biologically independent animals. **b** Log-rank test finds the three liver abscess-borne strains (Pa3$_{LS}$, Pa5$_{LR}$, and Pa6$_{LR}$) significantly more virulent than the bloodborne strains (Pa1$_{BS}$, Pa2$_{BR}$, Pa4$_{BR}$, and Pa7$_{BS}$): $p$ value = 0.0011. **c** Conversely, Log-rank test shows no significant difference in virulence when comparing the three phage-susceptible strains (Pa1$_{BS}$, Pa3$_{LS}$, Pa7$_{BS}$) with the four phage-resistant strains (Pa2$_{BR}$, Pa4$_{BR}$, Pa5$_{LR}$, Pa6$_{LR}$): $p$ value = 0.7552. **d** The absorbance at 570 nm was measured to evaluate the viability of HeLa cells. The highly virulent *P. aeruginosa* strain PA14 was used as a positive control whereas HeLa cells without the addition of a *P. aeruginosa* culture served as a negative control. A connected letter report was created to pairwise compare all isolates and controls (two-sided Student's *t* test, $p$ value = 0.05, $n = 6$ biologically independent samples, no adjustment made for multiple comparisons). Pa1$_{BS}$ was found to not significantly reduce the viability of the cells, compared to the negative control. Box-plots elements: center line, median; box limits, upper and lower quartiles; whiskers, 1.5× interquartile range; points, individual values. Source data are provided as a Source Data file.

cells. These human cells were incubated together with the *P. aeruginosa* isolates for 24 h, after which HeLa cell viability was assessed (Fig. 3.d).

No correlation was found between BPR phenotype and prolonged survival of the larvae or virulence towards HeLa cells (Fig. 3a, c, d).

Moreover, no correlation was observed between the data of the two virulence assays, although both assessments revealed Pa1$_{BS}$ as the least virulent isolate. However, we consider that this does not necessarily rule out the occurrence of PIVT in the child. Indeed, results from our assay potentially suffer from survivor bias, since they can only be

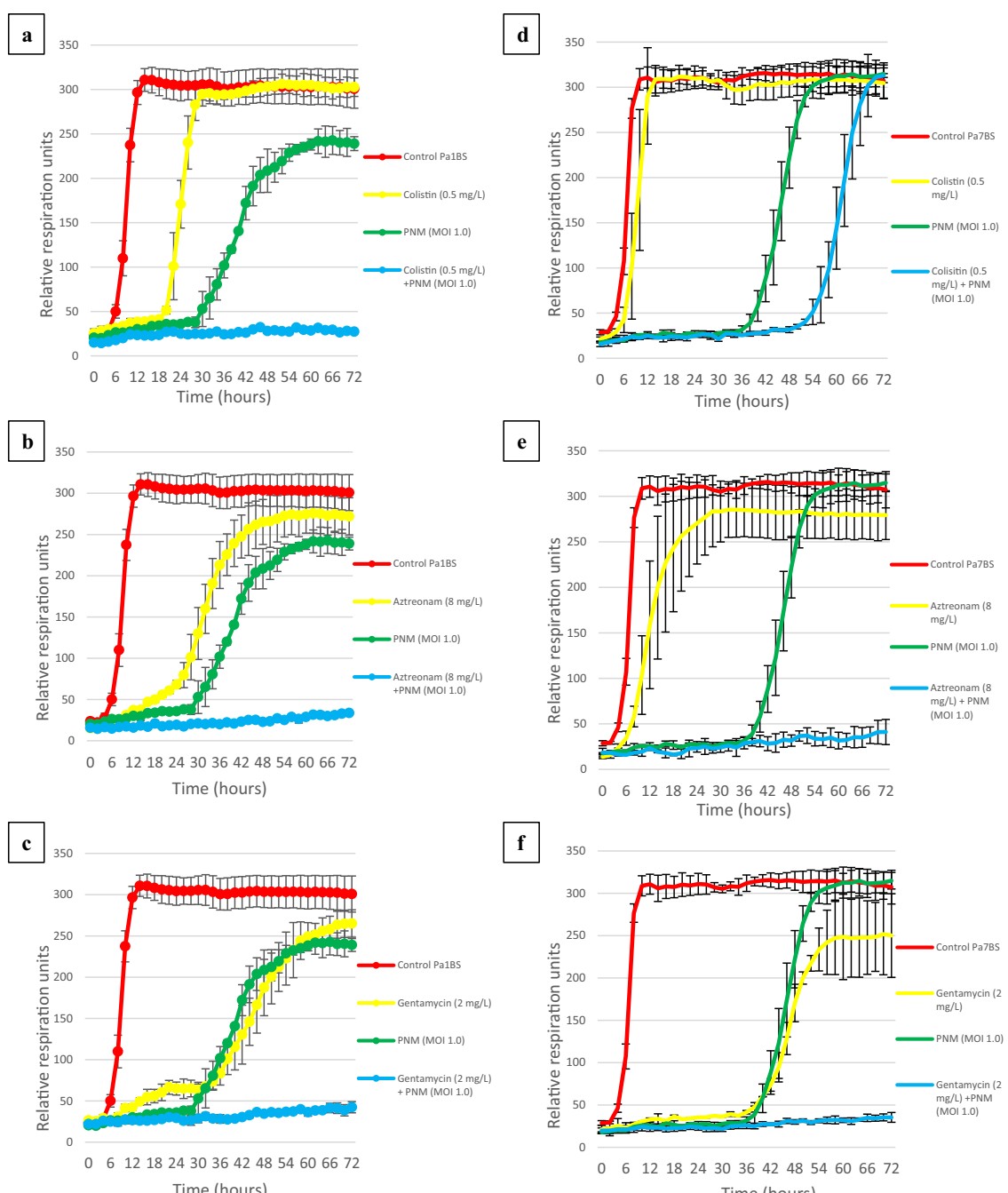

**Fig. 4 | Phage-antibiotic synergy assay results.** Activities of phage PNM (at multiplicity of infection [MOI] 1.0) and **a/d** colistin (0.5 mg/L), **b/e** aztreonam (8 mg/L), and **c/f** gentamycin (2 mg/L), as well as combinations of phage PNM and these antibiotics against *Pseudomonas aeruginosa* isolate $Pa1_{BS}$ (**a–c**) and $Pa7_{BS}$ (**d–f**) were determined using an OmniLog® system. Bacterial proliferation is presented through relative units of cellular respiration. Efficacious phages, antibiotics, and combinations thereof, suppress bacterial proliferation. Results are presented as mean values of three experiments (biological replicates) with error bars representing the standard deviations of the means. Source data are provided as a Source Data file.

based on the phage-resistant isolates that we managed to retrieve and culture from the child's biological samples. Such isolates are the ones that managed to escape both phage lysis and immune system clearance, but their bioburden could ultimately be too low to sustain or re-establish proper infection in the child. Such isolates could be considered "remnant" colonizers, fallaciously representing a bacterial population that was largely decimated by a combination of phage lysis and PIVT.

Based on the *Gm* assay, it should be noted that three isolates appeared significantly more virulent ($Pa3_{LS}$, $Pa5_{LR}$, and $Pa6_{LR}$) (*p* value = 0.0011) (Fig. 3b). According to our WGS analysis, these

isolates do not seem to share chromosomic characteristics that none of the other four strains ($Pa1_{BS}$, $Pa2_{BR}$, $Pa4_{BR}$, and $Pa7_{BS}$) harbor. This does not rule out the possibility of a significantly higher virulence in the first three isolates, since the spontaneous generation of phenotypic, non-genetically-mediated heterogenicity regarding such characteristics within a clonal bacterial population is a known phenomenon[19]. However, we noticed that these three isolates were the ones retrieved from liver abscess pus, whereas the other less virulent isolates were bloodborne. While this correlation between liver origin and virulence of the isolate in *Gm* could be coincidental, it could also provide a base for another putative explanation of PIVT in this model.

Indeed, PT's scheme of administration in our case was mainly intravenous. This makes it safe to assume that bacteria in the bloodstream were significantly more exposed to phages than bacteria in the liver abscesses. The lower virulence in *Gm* of the bloodborne "bacterial subpopulation" then correlates with its higher exposition to phages, whereas the lesser exposed liver subpopulation exhibits higher virulence. Under this paradigm, PIVT could be seen as a "subpopulation-scale" mechanism, where the correlation between the intensity of exposition to phages and virulence loss is not dependent on the phage-susceptibility of some specific bacterial individuals in these subpopulations. It should be noted, however, that the results of the subsequent HeLa cell assay did not reproduce this clustering.

Finally, we should also stress that we were unable to investigate a key component of the co-evolving host-parasite duet: the phages. Unfortunately, biological samples that would allow re-isolation of co-evolved phages from the patient's bloodstream or liver abscesses were not kept. As a result, the BPR phenotype is based on the phage clone that was initially used in PT, and which might have evolved in vivo to re-infect the emerging phage-resistant *P. aeruginosa* isolates.

## Phage-antibiotic synergy

Even though all PT randomized controlled trials that have been performed to date evaluated defined phage products as stand-alone therapies[20], PAS is increasingly being reported in the literature[21], where it is linked to the enhanced bacterial killing, increased penetration into biofilms, and decreased selection of antibiotic- or phage-resistant clones. We investigated the possible in vitro PAS between phage PNM and sub-optimal concentrations of colistin (0.5 mg/L), aztreonam (8 mg/L), and gentamycin (2 mg/L), which were administered in combination with PT (Fig. 1). Using an OmniLog® system we showed a clear in vitro synergistic activity for phage PNM and any of the three relevant antibiotics against *P. aeruginosa* isolate Pa1$_{BS}$ (Fig. 4a–c). Similar PAS was observed for isolate Pa7$_{BS}$, although colistin showed slightly less intense synergistic properties (Fig. 4d–f).

## Phage immune neutralization

Phages are able to elicit specific immune reactions in humans that could potentially eliminate the phages and impact PT efficacy[22]. As early as the late 1930's, it was assumed that the appearance of anti-phage antibodies typically could cause the failure of PT, and therefore the application of the same phage(s) for a period longer than two weeks was rarely prescribed in the former Soviet Union[23]. Recent reviews indicate that the presence of phage-specific antibodies can interfere with therapeutic efficacy, though admitting that such phenomenon has not yet been systematically addressed in patients[24]. A study in 20 patients with *S. aureus* infections revealed that PT induces significant anti-phage IgM and, particularly, IgG titers[25]. However, the deteriorating effect of such antibodies on PT's efficacy and on chances of clinical success seems highly dependent on the phages' administration route, their effect being seemingly very limited in cases of topical or digestive applications[22]. Moreover, the induction of phage-specific antibodies seems irrelevant for PT of acute infections, because antibodies are formed after the phages have performed their antibacterial effect. In contrast, phage-specific antibodies are more relevant in long-term PT of chronic infections or when repeated treatments with the same phages boost the humoral immune response. This was recently confirmed in a case study: an immunocompetent patient with *Mycobacterium abscessus* lung infection was treated for six months with a three-phage cocktail administered intravenously, and the emergence of a potent IgM and IgG neutralizing antibody response to the phages after one month of PT was associated with limited therapeutic efficacy after two months of PT[26].

We assessed the PIN capacity of ten serum samples, one taken prior to PT initiation (control), and the other nine at various time points during and after PT. Phage lytic activity (titer) was determined by double-agar overlay plaque assay, before and after incubation with the patient's serum samples. This assay thus indirectly assesses the PIN effect of serum samples against a given phage strain. The rationale behind this assay is that it is understood that only phage-neutralizing immunoglobulins in a serum sample could be responsible for a time-dependent loss of phage titer. As such, this method does not directly detect specific antibodies against specific phage antigens. Over the entire 51-day period of IV PT covered by the serum samples, no PIN effect was observed against *P. aeruginosa* phages PNM and 14-1. In contrast, *S. aureus* phage ISP, however, was shown to elicit a PIN effect. The chronological evolution of the PIN reaction against ISP is represented in Fig. 5. The PIN effect against phage ISP emerged during the fifth week of PT and reached a steady maximum during the sixth week. An additional serum sample collected approximately one-year after DDLT showed that the PIN reaction had disappeared by then.

In our experience, the serum PIN pattern expressed by this child is atypical. Indeed, no PIN activity was detected against phages PNM and 14-1 during or after a very long IV application of these phages, whereas unpublished cases did detect potent PIN against PNM and 14-1, already starting significantly during the second week of PT. This could suggest that, at least in this case, this atypical pattern was not related to phage-dependent factors but rather to host-dependent factors, such as the persistent immunosuppression endured by the patient during the entire duration of PT[27]. Indeed, repeated serum gamma-globulin and blood lymphocyte quantifications indicate that the child endured a sustained lymphopenia whose degree of severity varied over time, with the most severe stages being observed during the onset of bacteremic episodes, such as the *P. aeruginosa* bacteremia starting four days before PT initiation (Fig. 5c, d). The child's general immune-incompetence, and his lymphopenia in particular, are most probably multifactorial. A considerable body of work has documented a constitutional immune immaturity in children, especially of the adaptive mechanisms, including an immaturity of T-independent B-cell activation up to the age of five years[28]. In addition, since the child was treated with corticosteroids and tacrolimus since his first liver transplantation, the drug-induced component of the lymphopenia and suspected lymphocytic malfunction should not be underestimated. The most severe stages of lymphopenia (reaching lows of <300 cells/μL) are likely linked to the bacterial sepsis episodes and associated apoptosis-induced lymphopenia[29]. The fact that phage PNM, which showed an in vitro lytic activity against the patient's *P. aeruginosa* strain, did not elicit a PIN reaction in this immunosuppressed child was fortunate, as it allowed a prolonged (86 days) therapeutic and prophylactic IV application of BFC1. The observation that phage ISP was able to elicit PIN in this immunocompromised patient does not seem to be due to especially potent immunomodulatory properties of phage ISP compared to those of phage 14-1 or phage PNM[30]. We did not assess whether higher intrinsic immunogenicity of some surface epitopes of phage ISP, like specific glycoproteins, might explain this phenomenon[31].

In conclusion, long-term (86 days) IV PT was shown to be safe in this vulnerable young child with uncontrolled *P. aeruginosa* bacteremia, advanced biliary disease, and immunosuppression. PT initiation was associated with a marked and sustained improvement of the child's clinical condition, and resulted a few days later in the definitive eradication of *P. aeruginosa* from the bloodstream. This favorable progression enabled a dearly needed liver re-transplantation. Two years later, the child remains well and has not suffered any new infection. This is the longest reported IV PT in a child and, to our knowledge, also the first time that PT instillation was performed during the anhepatic phase of liver transplantation surgery. It is known that BPR, PAS, and PIN can occur and can have an impact on PT. We observed BPR against native phage PNM in four out of seven *P. aeruginosa* isolates retrieved from both the bloodstream and liver abscess

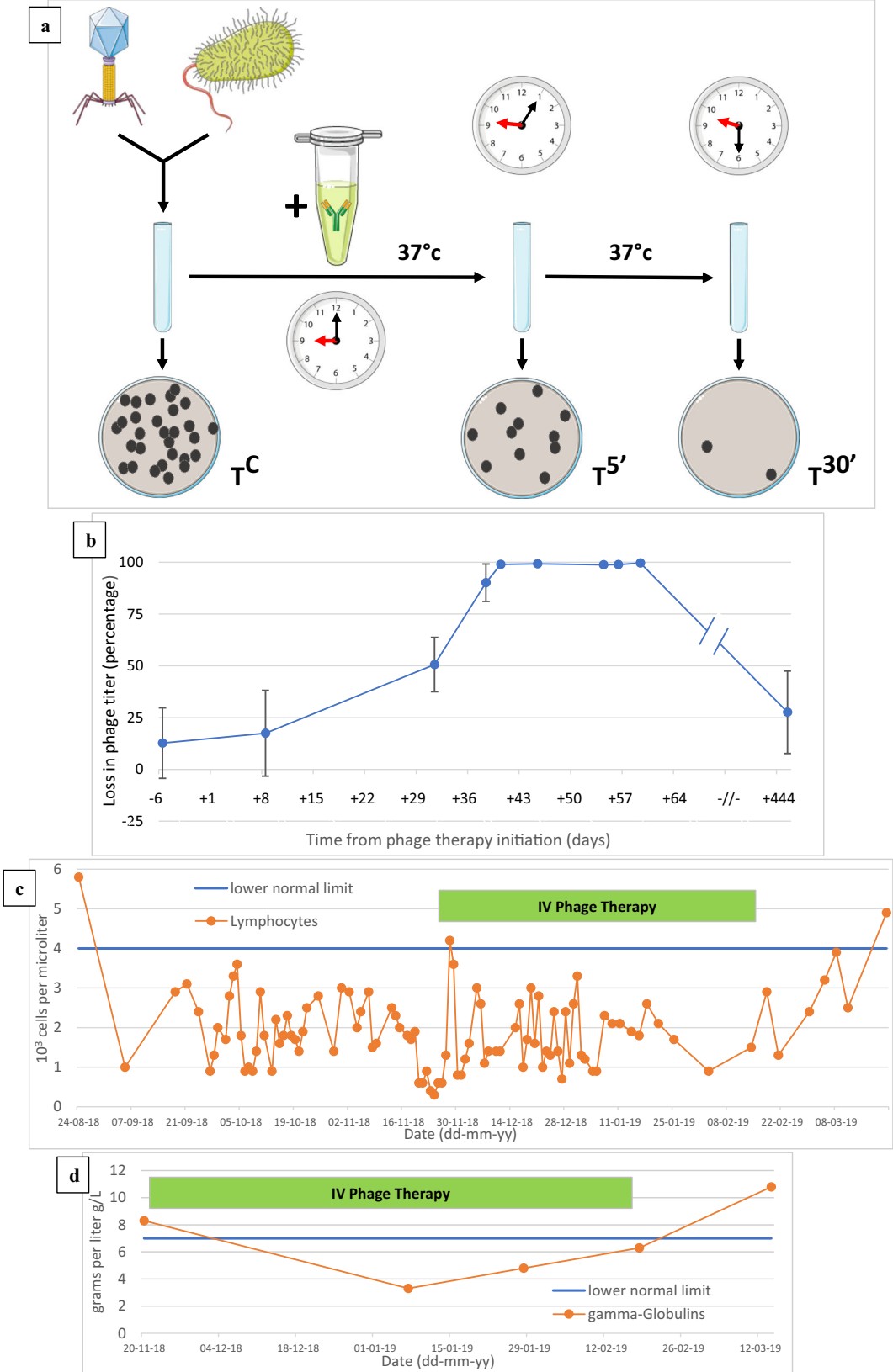

pus. Neither the use of a *Galleria mellonella* virulence model nor the use of a HeLa cell virulence model allowed us to illustrate PIVT directly between the phage-susceptible and phage-resistant bacterial sub-populations, but the *Galleria mellonella* assay did show a significantly higher virulence of liver-borne isolates compared to bloodborne

isolates. Whether this difference is linked to a difference in the degree of exposure to phages in these bacterial subpopulations is not certain, and this clustering was not observed in the HeLa cell virulence assay. The results of these assays possibly suffer from survivor bias. Choosing a valuable model on a case-by-case basis to investigate possible PIVT is

**Fig. 5 | Phage immune neutralization (PIN) assay. a** Schematic overview of the assay's methodology ($T^C$: control titer; $T^5$: titer after 5 min of serum-phage co-incubation; $T^{30}$: titer after 30 min of serum-phage co-incubation). **b** Chronological phage immune neutralization (PIN) activity against phage ISP of ten patient sera collected before, during and after phage therapy. The evolution over time of the PIN activity against phage ISP is shown. Serum PIN activity is shown as % phage titer loss (compared to a pre-PT control serum) after incubation of phage ISP with sequential serum samples for 30 min. PIN activity appeared during the fifth week of PT initiation and had disappeared in a one-year post re-transplantation serum sample. Data are presented as mean values with error bars representing the standard deviation of the means. Each serum sample was tested on at least three independent occasions: six times for the 24-12-18 sample, four times for the 07-01-19 and 18-01-19 samples, and three times for all the other samples. **c, d** Persistent biological immunosuppression over the course of PT is illustrated through repeated quantification of the patient's blood lymphocytes (normal values $4.0-10.5 \times 10^3$ cells/microliter) and gamma-globulins (normal values $7.0-15.0$ grams/liter). Source data are provided as a Source Data file.

an experimental challenge that warrants further research, and should probably be based on bacterial species, phage receptor and the contribution of this receptor to the bacterial virulence phenotype. In vitro PAS was observed for phage PNM in combination with any of the three antipseudomonal antibiotics that were administered simultaneously: colistin, aztreonam, and gentamycin. Finally, PIN reaction was considered incomplete, as it fortunately did not develop against anti-*P. aeruginosa* phages PNM and 14-1. This is likely due to patient-dependent factors including a multifactorial severe lymphopenia during the entire duration of PT. In-patient phage pharmacokinetics were however not directly measured, which mitigates the accurate evaluation of PIN and PAS contribution to the patient's outcome, as these mechanisms affect or rely on in-patient phage titers. In the light of the antimicrobial resistance crisis and the paucity of novel antibiotics, PT represents a noteworthy additional tool, in synergy with last-resort antibiotics, to treat challenging infections in patients, including organ transplant recipients and children.

## Methods

### Ethical approval and consent
Informed consent was obtained from both the patient's parents. Ethics Committee approval to initiate phage therapy in accordance to the article 37 of the Declaration of Helsinki was delivered by the Comité d'Éthique hospitalo-facultaire Cliniques universitaires Saint-Luc/UCLouvain on November 23rd, 2018.

### Antibiotic susceptibility
At the QAMH, antibiotic susceptibilities of the *P. aeruginosa* isolates were ascertained using the VITEK 2 system (bioMérieux). At Saint-Luc University Hospital, antibiotic susceptibilities were ascertained through disk diffusion method using Adagio reading technology (Bio-Rad, Marnes-la-Coquette, France) and through automated microdilutions (Phoenix, Becton Dickinson, Franklin Lakes, NJ, USA). Categorization (therapeutic interpretation) of Minimum Inhibitory Concentrations (MICs) were based on European Committee on Antimicrobial Susceptibility Testing (EUCAST) guidelines.

### Genome sequencing and analysis
The genomes of the seven *P. aeruginosa* strains were sequenced as previously described[10]. Total genomic DNA (gDNA) was extracted from the isolates with the DNeasy UltraClean Microbial kit (Qiagen) according to the manufacturer's instructions. The gDNA was subsequently prepared for Illumina sequencing using the Nextera Flex (Illumina) and sequenced on an Illumina MiniSeq machine using a paired-end approach ($2 \times 150$ bp). In addition, the gDNA was also prepared for long-read sequencing using the Rapid barcoding kit SQK-RBK004 (Oxford Nanopore Technology) and sequenced on a MinION equipped with a R9.4.1 flowcell (Oxford Nanopore Technology). The quality of the Illumina sequencing data was assessed using FastQC v0.11.9 and trimmomatic[32] v0.39 for adapter clipping, quality trimming (LEADING:3 TRAILING:3 SLIDINGWINDOW:4:15), and minimum length exclusion (>50 bp). Quality of the Nanopore reads was assessed using Nanoplot[33] v1.30, and Porechop v0.2.3 was used for barcode clipping and NanoFilt[33] v2.7.1 for filtering on quality ($Q > 8$) and length (>500 bp). The genomes were reconstructed using the

hybrid assembler Unicycler[34] v0.4.8. SNP calling was done using snippy v4.6.0 against the NCBI PGAP[35] annotation of isolate $Pa3_{LS}$. Large structural variations were inspected using ngmlr v0.2.7 and sniffles[36] v1.0.12. The assemblies were visually inspected using Bandage[37] v0.8.1. Serotyping and sequence typing were performed using PAst[38] (accessed in April 2020) and the mlst software v2.19.0. Further functional annotation, antibiotics resistance genes, prophage elements, and genomic islands were annotated using respectively eggNOG-mapper[39] v2, abricate v1, PHASTER[40] (accessed in April 2020), and islandviewer[41] 4. All sequencing data generated in this study has been deposited in the NCBI BioProject database accession PRJNA776240.

### *Galleria mellonella* virulence assay
The seven retained *P. aeruginosa* isolates ($Pa1_{BS}$ to $Pa7_{BS}$) were cultured and then diluted in lysogeny broth (LB; 10 g/L tryptone (Neogen, Lansing, Michigan, USA), 5 g/L yeast extract (Neogen), 10 g/L NaCl (Acros Organics, Geel, Belgium)) until reaching an optical density (at wavelength 600 nm) of 0.25–0.35. Each bacterial suspension was then centrifuged and the bacterial pellet was resuspended in PBS (self-made, composition: 137 mM NaCl, 2.7 mM KCl, 10 mM $Na_2HPO_4$, 1.8 mM $KH_2PO_4$, all components from Sigma-Aldrich, Merck KGaA group, 64293 Darmstad, Germany). The PBS suspension was diluted ($5 \times 10^{-6}$ dilution factor) in PBS and 20 µL, representing an inoculum of approximately ten colony forming units (cfu), was injected in a *G. mellonella* larva. The larvae were all of the same age and were separated into eight batches of larvae (one batch for each of the seven *P. aeruginosa* isolates, one for the PBS control. The assay was first calibrated using batches of five larvae, to standardize methodology of injection and estimate intra-batch variance as well as optimal timeframe for follow-up. The assay was then performed with batches of ten larvae and each batch was standardized for weight. Each batch of ten *G. mellonella* larvae was inoculated with one of the seven *P. aeruginosa* isolates, and one batch was inoculated with 20 µL of PBS (control). Intrahemocoelic injection was performed at the basis of the hindmost left pro-leg of each larva, using thin intradermal needles (Micro-fine + insulin U100 0.3 ml syringe, Becton Dickinson, Franklin Lakes, New Jersey, USA). The *G. mellonella* larvae were stored in petri dishes (one dish per batch) and kept in the dark at 37 °C. Each larva was inspected visually every hour for 16 h to assess its activity score. The activity score ranges from 0 to 9 points (Supplementary Fig. 4).

### HeLa cell viability assay
The virulence of the *P. aeruginosa* isolates against HeLa cells was assessed via a cell viability assay, based on the protocol described in Hernandez–Padilla et al.[42]. Briefly, human cells were cultured in Dulbecco's modified eagle medium (DMEM; Thermo Fisher Scientific) supplemented with 10% fetal bovine serum (FBS; Thermo Fisher Scientific) and 1% antibiotic–antimycotic solution (Sigma Aldrich) (referred to as DMEM+10) and incubated at 37 °C with 5% $CO_2$. When confluency was reached, $3 \times 10^4$ cells were seeded in a 96-well plate and incubated overnight. Next, $10^6$ bacterial cells in DMEM + 10 lacking phenol red were collected based on optical density ($OD_{600\ nm}$ of 0.5 corresponds to $2 \times 10^8$ cells) and added to the HeLa cells in a final

volume of 200 μL. The following day, the bacterial cells were removed and the medium was refreshed. According to the protocol of the manufacturer, 10 μL MTT (Sigma Aldrich) was added to the cells and incubated for four hours. Finally, 100 μL solubilization buffer (Sigma Aldrich) was added to disolve the crystals. The viability of the cells was assessed by their ability to cleave MTT (tetrazolium dye) via mitochondrial dehydrogenases. By cleavage of MTT, formazan is formed and is measured spectrophotometrically at 570 nm in a microplate reader (CLARIOstar Plus, BMG Labtech). The experiment was performed with six technological replicates. A Student's $t$ test with the non-infected HeLa cells as control group was executed to pairwise compare the different treatment groups (significance level $p < 0.05$) using the JMP v16 software (SAS Statistics, Cary, North Carolina, USA).

### Bacterial growth kinetics

Bacterial cellular respiration was measured using the OmniLog system (Biolog, Hayward, CA, USA). Data were analyzed with OmniLog data Analysis Software (v1.7). Growth kinetics of all seven isolates were first measured in the absence of any phage or antibiotic. Then, to investigate PAS, the growth kinetics of *P. aeruginosa* isolates Pa1$_{BS}$ and Pa7$_{BS}$ were assessed in the presence of phage PNM (at MOI 1.0), colistin (0.5 mg/L), aztreonam (8 mg/L), and gentamycin (2 mg/L), and combinations of phage PNM and these antibiotics. Experiments were done in 96-well plates in a final volume of 200 μl of LB supplemented with 100 times diluted tetrazolium dye mix A, according to the manufacturer's instructions. Bacterial cells were added at a concentration of $10^5$ cfu/well, calculated based on optical density (OD, at 600 nm) measurements (with an OD of 0.5 corresponding to $4 \times 10^8$ cfu/ml, on average), which were validated using a classical plate culture method. The titer of phage PNM was also confirmed after each experiment using the classical double-agar overlay method[43]. Plates were incubated at 37 °C for 72 h and a possible reduction (causing a color change) of the tetrazolium dye due to bacterial respiration (during growth) was monitored and recorded every 15 min by the Omnilog system. Efficacious phages, antibiotics, or combinations thereof, suppress bacterial growth (respiration). Experiments were performed in triplicate (biological replicates).

### PIN assay

The PIN activity of the patient's serum samples was determined according to Adams[43] with some modifications. Phage lytic activity (titer) was determined by double-agar overlay plaque assay before and after incubation of the phages with the patient's serum samples. We used a selection of ten serum samples that were harvested from the toddler over a 2-month period, from one week before PT initiation to day 51 of continuous IV PT. Because this assay assesses the PIN capacity of a serum sample against one phage strain, the test was performed for each of the three phage components of BFC1. PIN intensity is correlated to the time-dependent loss of phage titer after incubation with sera. The phage solution is first titered using its reference host bacterial strain using a standard double-agar overlay titration method, and this titer is used as a control measurement (Tc). Blood was allowed to clot for a minimum of 30 min in a vertical position and then centrifuged at room temperature in a swinging bucket rotor for 10 min at 2000×$g$. Serum samples were stored at −80 °C ± 5 °C. To assess the effect of the sequential serum samples on phage lytic activity, 0.9 ml of the diluted sera (1:100 in self-made normal saline, NaCl 0.9%) were mixed with 0.1 ml of the phage at a concentration of $2 \times 10^7$ pfu/ml and incubated at 37 °C. After 5 min and after 30 min of incubation, the phage titer of the phage-serum mixture is determined using the same double-agar overlay titration method, generating titers T5′ and T30′, respectively (Fig. 5a). In this method, a significant loss of titer both between TC and T5′ and between T5′ and T30′ is highly indicative for the presence of phage-neutralizing antibodies in the tested serum.

Conversely, the absence of significant differences between titers TC, T5′, and T30′ suggests an absence of phage-neutralizing antibodies. For each serum sample, the PIN assay was performed between three and six times to generate robust mean results.

### Statistics and reproducibility

No data were excluded from the analyzes. Each assay is based on at least three independent measurements ($n = 3$–10 depending to the assay, as described in the main text and in the respective Figure legends). No statistical method was used to predetermine sample sizes. Investigators were blinded to allocation and outcome assessment during the *Galleria mellonella* and HeLa cells assays. The *Galleria mellonella* larvae were standardized for weight and randomly allocated to any of the 8 experimental groups. Randomization was not considered applicable to the other experiments.

### Reporting summary

Further information on research design is available in the Nature Research Reporting Summary linked to this article.

## Data availability

All sequencing data generated in this study have been deposited in the NCBI BioProject database accession PRJNA776240. Source data are provided with this paper.

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

## Acknowledgements

Phages PNM and ISP were originally obtained from Eliava Institute, Tbilisi, Georgia. Phage 14-1 was received from Russia via Professor Krylov, http://eng.genetika.ru/, RIGSIM, current abbreviation. We thank A. Buckling for his insight regarding *Galleria mellonella* assays and PIVTs. We thank L. Putzeys for her help regarding *Galleria mellonella* handling and assay protocol. We thank F. Van Bambeke for her technical support regarding PAS assays. We thank L. Belkhir for her insight on bloodborne and abscess-borne bacterial subpopulations. We thank S. De Soir, J. Griselain, G. Steurs, and C. Cochez of Laboratory for Molecular and Cellular Technology for their technical support in phage and bacterial culture, PIN assays, and PAS assays. B. Van Nieuwenhuyse is a FRIA Grant Holder (grant number: FC 36489) of the Fonds de la Recherche Scientifique-FNRS. C. Lood is supported by a Ph.D. fellowship from FWO Vlaanderen (1S64720N) and a PDM grant from KU Leuven (PDMT2/21/038). M. Merabishvili is supported by grant HFM 19-12 from the Royal Higher Institute for Defense. M. Merabishvili, R. Lavigne, and J.-P. Pirnay is member of the 'PhageBiotics' research community of the FWO Vlaanderen.

## Author contributions

B.V.N., D.V.D.L., O.C., C.L., J.W., R.L., K.S., H.R.-V., S.D., M.M., P.S., and J.-P.P. contributed to data collection, analysis, interpretation, and writing. C.D.M. contributed to surgical management, clinical description, and writing. D.V.D.L. E.S., X.S., and I.S. contributed to medical management, clinical description, and writing. S.D. and P.S. contributed to phage supply, phage therapy administration protocol, medical management, and writing. C.L., J.W., and R.L. contributed to genomic data collection, assembly, and analysis. H.R.-V. contributed to microbiological analysis regarding antimicrobial susceptibility. B.V.N. and M.M. contributed to phage immune neutralization assays. B.V.N., J.W., and J.-P.P. contributed to *Galleria mellonella* assays. K.S. and J.P. contributed to the HeLa virulence assay. B.V.N. and M.M. contributed to phage-antibiotic synergy assays.

## Competing interests

The authors declare no competing interest.

## Additional information

**Brieuc Van Nieuwenhuyse** [1] [✉]**, Dimitri Van der Linden**[1,2]**, Olga Chatzis**[2]**, Cédric Lood** [3,4]**, Jeroen Wagemans** [3]**, Rob Lavigne** [3]**, Kaat Schroven** [3]**, Jan Paeshuyse**[5]**, Catherine de Magnée**[6]**, Etienne Sokal**[7]**, Xavier Stéphenne**[7]**, Isabelle Scheers**[7]**, Hector Rodriguez-Villalobos** [8]**, Sarah Djebara**[9]**, Maya Merabishvili**[10]**, Patrick Soentjens**[9,11] **& Jean-Paul Pirnay** [10]

[1]Institute of Experimental and Clinical Research, Pediatric Department (IREC/PEDI), Université catholique de Louvain - UCLouvain, Brussels, Belgium. [2]Pediatric Infectious Diseases, General Pediatrics Department, Cliniques universitaires Saint-Luc, Université catholique de Louvain - UCLouvain, Brussels, Belgium. [3]Department of Biosystems, Laboratory of Gene Technology, KU Leuven, Leuven, Belgium. [4]Department of Microbial and Molecular Systems, Centre of Microbial and Plant Genetics, Laboratory of Computational Systems Biology, KU Leuven, Leuven, Belgium. [5]Department of Biosystems, Laboratory of Host Pathogen Interactions, KU Leuven, Leuven, Belgium. [6]Pediatric and Transplantation Surgery, Cliniques universitaires Saint-Luc, Brussels, Belgium. [7]Pediatric Hepatology and Gastroenterology, Cliniques universitaires Saint-Luc, Brussels, Belgium. [8]Department of Microbiology, Cliniques universitaires Saint-Luc / Université catholique de Louvain - UCLouvain, Brussels, Belgium. [9]Center for Infectious Diseases, Queen Astrid Military Hospital, Brussels, Belgium. [10]Laboratory for Molecular and Cellular Technology, Queen Astrid Military Hospital, Brussels, Belgium. [11]Department of Clinical Sciences, Institute of Tropical Medicine, Antwerp, Belgium. ✉e-mail: brieuc.vannieuwenhuyse@uclouvain.be

