## [Peer review file · Nature Communications]

REVIEWER COMMENTS

Reviewer #1 (Remarks to the Author):

General comments:

Manuscript "Bacteriophage-antibiotic combination therapy against extensively drug-resistant *Pseudomonas aeruginosa* infection to allow liver transplantation in a toddler" by Van Nieuwenhuysen et al is a research article presenting the case of a child suffering from sepsis caused by a multidrug-resistant strain of *P. aeruginosa* after liver transplantation who received combined phage and antibiotic therapy. The work takes into account microbiological issues in detail, in particular studies of the susceptibility of bacterial isolates to phages and antibiotics as well as strain genetic similarity. The authors try to answer the question /explain the mechanism of variability among the isolated strains. The work presented for evaluation has a special meaning as it concerns the use of phage therapy combined with antibiotic in a transplanted child, and its results may have a fundamental contribution to the development of phage therapy, its availability and widespread therapeutic application both as a confirmation of their efficacy and of the synergy between phages and antibiotics. The work has a significant contribution to the development of the contemporary medicine. It is also well organized and comprehensively described study. In my opinion the experiments conducted in the presented manuscript were designed appropriately and the manuscript has adequate references to related and previous work. Overall the article is well-organized, and well explained with up to dated reference. The obtained results are a real success.

Detailed comments:

1. Unfortunately, due to the size of the Figure 1, the details of the medical procedures used are not clearly legible. Is it possible for Figure 1 to be in better resolution?
2. I understand that the preparation used contained purified phages? By what method were the phages purified, what was the level of bacterial endotoxins in the administered preparation? Please provide this information.
3. Line 110: What do the authors mean "without any change of phage cocktail? Activity?"
4. Line 365: Please remove the extra "of the" from the sentence.
5. What do the authors know about the sequences of the applied phages and their contents (the presence of genes encoding integrase, antibiotic resistance genes, toxin genes)?

Reviewer #2 (Remarks to the Author):

Summary

The paper “Bacteriophage-antibiotic combination therapy against extensively drug-resistant *Pseudomonas aeruginosa* infection to allow liver transplantation in a toddler” is a clinical case study about a child (1 year old) that received phage therapy and concomitant antibiotics which led to the eradication of an XDR *Pseudomonas aeruginosa* infection.

This was a noteworthy study because the authors took a successful phage therapy clinical case study a step further by trying to investigate the full contribution of phage to the favorable outcome. The authors tried to investigate further through in vitro and in vivo work to determine bacterial phage resistance (BPR), phage-induced virulence tradeoffs (PIVT), phage-antibiotic synergy (PAS), and phage immune neutralization (PIN). The authors used inventive methods including the *Galleria mellonella* (Gm) to investigate PIVT and the OMNILOG system to investigate PAS. These are useful tools and methods for the scientific community.

A few minor suggestions for changes and further elaboration to support the conclusions are needed and included below.

Major Comments

Related to phage-induced virulence tradeoffs (PIVT)

1. The authors noted a significant difference in virulence as determined using Gm between the recovered blood isolates and liver isolates. However, on pg 10, lns 243 – 244 it states “these isolates do not seem to share chromosomic characteristics that none of the other four strains harbor”.

a. Can you explain this further as to what these differences could be that allow the bacteria to be more virulent, if not chromosomal (in the text)? Are there differences in bacterial growth between the isolates and that is why they have differences in virulence? Was this tested?

2. Pg 10, Ins 242 – 254 . I was confused by the reasoning in this paragraph. The liver isolates and blood isolates share the same chromosomal characteristics, but the blood isolates were more exposed to the phage? Would this suggest that they were both exposed to the phage if similar phage resistance mutations were found in the liver isolates?

Phage Immune Neutralization (PIN):

1. I was confused by the reasoning pg 14 In 328 – 330. Serum PIN was determined with ISP and not for PNM and 14-1. These are three different phages. Would this suggest this is phage-dependent?

2. ISP eliciting an immune response in an immunosuppressed patient seems to be one of the more interesting aspects of this study. Could ISP have stimulated immunity and what contribution could that have made to the success phage therapy? Could you talk about this further in the text?

Minor changes

1. Please include page numbers for the future.

2. Pg 2, Figure 1. This figure is difficult to read. Not sure if this is due to lower resolution. Please use higher resolution figures or larger, easier to read font (darker colors).

3. Pg 3, Ins 93-95 In the text it reads “...two transient *P. aeruginosa* translocations from liver abscesses to the bloodstream were observed on day 27 and day 30 after PT initiation...” however, this event is not noted on the timeline in Figure 1. It seems important, although transient, that an infection occurred during phage treatment. Please include the event on the timeline.

4. Pg 12, In 309. Should it say “data not shown” after this sentence? Is this data available?

5. Pg 16, Ins 405 and 417. Missing information for Manufacturer, city and country.

Reviewer #3 (Remarks to the Author):

Summary

The manuscript describes a child with extreme immune ablation who was treated for a liver abscess with antibiotics and phages. The one phage with lytic activity in vitro also exhibited antibiotic-synergistic killing of isolates from the blood that appeared not to have been forced to sacrifice a known virulence factor (type 4 pilin) to escape phage predation.

It is inferred that the invasive subpopulations detected as the bacteraemic isolates were extinguished by the synergistic combination therapy, leaving only less invasive isolates that were detected in the abscess, which was directly irrigated with phages. The fitness costs of phage-resistance in bacteria from the liver abscess are ill-defined, perhaps because the assays used did not depend primarily on the trade-off observed.

Antibiotic gene traffic appears to be independent of any selection pressure and is not relevant (see below).

The demonstrated lack of phage neutralisation might have ensured good delivery to site of action but phage kinetics were not directly measured. In fact, the dose given into the estimated blood volume is likely to yield a relatively low multiplicity of infection. The authors avoided this contentious area but it might have been mentioned briefly in Discussion – perhaps the demonstrated antibiotic synergy was particularly important in this case.

Some of the figures might be revised or omitted and some questions remain open. The work has been well executed on the material that was available. The impact of lack of neutralisation on phage kinetics and the virulence cost of the tradeoff of the (previously shown) receptor remain undefined.

Detailed comments

The case

About 2×10^7 pfu/kg were administered for $\sim 3/12$ in a child with *P. aeruginosa* sepsis

The phage cocktail was 2 antipseudomonal phage and a staph phage, only one of which (PNM) generated confluent lysis on solid media, at less than a third of the reference production strain (EOP 0.2-0.3) and this activity was only present against three of the seven isolates.

10^8 pfu (10^7 pfu/kg) were administered by slow infusion daily for a week, supplemented by direct instillation of the same amount into the liver abscess cavity.

10^8 pfu were administered daily into a blood volume of (say) 800mL, supplemented by intralesional injections of 10^7 pfu. The 10^8 pfu intravenous dose is diluted to $\sim 10^7$ in blood, at which concentration only semi-confluent plaques are evident in vitro, and only in isolates 1 (pre-dose), 3 (after first dose) and 7 (a week after starting). In the second week, IV dose was doubled and 2×10^8 (2×10^7 pfu/kg) was given on d66-d143 ie for about ten weeks

specific comments

Fig 1 is an excellent summary and overview of the clinical course and could be enlarged for easier cross-reference during the reading of the main text.

Fig 2 is unnecessary

Re Fig 3(c) (Table) - the antimicrobial activity is presented as three separately determined and partially contradictory results for vegetative phase organisms and the genomic and genetic analysis implies an active mobile gene pool with cassette-based genes variably present in isolates from the same population at the same site at the same time. The data imply that the gene pool mobility is completely independent of and unaffected by the beta-lactam and aminoglycoside antibiotics used.

Lines 168-170 seem to imply that this gene traffic is relevant but perhaps the authors may wish to consider the comments below.

Specifically - the integron-borne blaVIM (carbapenemase) gene cassettes, and the gene cassettes encoding aac 6' 29a and b which were first described with VIM-2 in *Pseudomonas*: (DOI: 10.1128/AAC.45.2.546-552.2001), all appear to be moving without regard to any selection pressure. The absence of the VIM-2 carbapenemase has no apparent effect on resistance to carbapenems, implying that the resistance is porin/ pump-mediated and, indeed, the finding of aztreonam susceptibility (the VIM carbapenemases being vulnerable to this synthetic monobactam) in all but isolates 2 and 6 implies that VIM-2 is not being selected at all. There is no annotated cephalosporinase such as PSE-2 identified in these but PDC-55, a class C type beta-lactamase normally found in *Pseudomonas*, is identified in all strains would be expected to hydrolyse aztreonam and might explain the carbapenem resistance phenotype in the presence of reduced permeability (eg an oprD lesion) or complementary efflux mechanism. Neither the ubiquitous (and in these isolates, persistent) aph 3'-1b (straA, also a very widespread class 1 integron gene cassette) nor the aac(6')-29a and 29b genes encode gentamicin resistance. The gene traffic described in Table /Fig 3(c) therefore just illustrate an active gene pool but not one that is convincingly influenced by any described selection pressure.

Lines 170-185. This is an interesting and important commentary – essentially, I think the authors are describing the simple trade-off of a type 4 pilus in a subpopulation within the liver abscess to avoid phage predation. We assume that the liver abscess is the primary site of infection and the finding of an 'entire' T4P expressing *P. aeruginosa* isolate in the blood at day 8 implies that this T4P tradeoff is a dominant subpopulation in the liver but not particularly successful as an invader. The known role of T4P in adhesion and biofilm initiation means that the choice of the galleria virulence model may not be ideal. Twitching motility assays might possibly be affected (although the implication from the genomic data is that the pilB rather than the pilT operon is impaired) but a growth competition assay or a virulence assay that does not depend on biofilm formation or pilin-mediated adhesion might be predicted to be inconclusive.

The discussion in lines 230-250 do not particularly address this and, even if an argument about non-specific fitness is to be made, a simple in vitro growth competition assay (between phage R and phage S) might be a better proof.

The phage-antibiotic synergy is nicely presented and are interesting, potentially important data. The convincing in vitro data imply that even a less effective phage and/or antibiotic might have impact when used in this way. Given the relatively unchanging phenotype over time and site for all three antibiotics, it

might have been useful to show whether this synergy was at all evident in other than the initial isolate, as the reader would expect this not to be the case in the liver but presumably still true in the blood on day 8 of phage therapy.

Finally, the findings from phage neutralisation assays are important even if unsurprising. The relevance is presumably that the lack of neutralisation means that viral delivery from blood is guaranteed over time and that immune complex-mediated disease is avoided. No viral kinetics were measured so this finding in a critically ill child with a completely ablated immune system bears little on the case.

The phage-antibiotic synergy may have been a saving grace in this clinical scenario but there are no drug or phage measurements in the blood available to directly show that. Nevertheless, this is an interesting and thought-provoking anecdote.

Reviewer #1 : response to comments

Dear Reviewer,

We thank you sincerely for your time and consideration in reviewing our manuscript. We took your remarks into consideration and believe this will strengthen our work. Please find a point-by-point response to your comments below.

Hoping this response brings you full satisfaction, we remain at your disposal for any further information.

Best regards,

Brieuc Van Nieuwenhuysse

1.1. Unfortunately, due to the size of the Figure 1, the details of the medical procedures used are not clearly legible. Is it possible for Figure 1 to be in better resolution?

→ The resolution of Figure 1 was increased.

1.2. I understand that the preparation used contained purified phages? By what method were the phages purified, what was the level of bacterial endotoxins in the administered preparation? Please provide this information.

→ We used *Bacteriofaag cocktail* batch 1.10 (BFC 1.10), which was purified by EndoTrap® affinity matrix system and then certified by Sciensano (the Belgian Scientific Institute of Public Health) to contain less than 5 EU/mL of endotoxins. **We added this precision to the revised manuscript.**

1.3. Line 110: What do the authors mean “without any change of phage cocktail? Activity?”

→ We deleted this part, which might have been unnecessary: we simply meant that the composition of phage cocktail BFC 1.10 did not change at any point during the case's progression (i.e. always the same three phages as described, no other phage added or deleted, ...).

1.4. Line 365: Please remove the extra "of the" from the sentence.

→ The extra "of the" was removed.

1.5. What do the authors know about the sequences of the applied phages and their contents (the presence of genes encoding integrase, antibiotic resistance genes, toxin genes)?

→ All three phages composing BFC 1.10 have been fully sequenced and characterized: None of the phages contained any known genetic determinants coding for lysogeny, toxins, or antibiotic resistance. All three phages are considered strictly virulent. Based on the genome analysis, Sciensano

(the Belgian Scientific Institute of Public Health) issued a “green” genomic passport, certifying the absence of any known genetic determinants that would make these phages unsuitable for therapeutic use. **We added this precision to the revised manuscript.** This was also described in the following reference: Merabishvili M, et al. Quality-controlled small-scale production of a well-defined bacteriophage cocktail for use in human clinical trials. PLoS One 4, e4944 (2009).

Reviewer #2 : response to comments

Dear Reviewer,

We thank you sincerely for your time and consideration in reviewing our manuscript. We took your remarks into consideration and believe this will strengthen our work. Please find a point-by-point response to your comments below.

Hoping this response brings you full satisfaction, we remain at your disposal for any further information.

Best regards,

Brieuc Van Nieuwenhuysse

Major

2.1. The authors noted a significant difference in virulence as determined using Gm between the recovered blood isolates and liver isolates. However, on pg 10, lns 243 – 244 it states “these isolates do not seem to share chromosomal characteristics that none of the other four strains harbor”.

- Can you explain this further as to what these differences could be that allow the bacteria to be more virulent, if not chromosomal (in the text)?

→ The spontaneous generation of phenotypic heterogeneity, with respect to characteristics that impact virulence, within a clonal bacterial population (non-genetic diversity) is a known phenomenon. This heterogeneity can arise from specific genetic architectures amplifying stochastic fluctuations in factors affecting gene expression, which drives variations that can affect the outcome of infection (e.g. the emergence of persister cells). **We will mention this in the revised manuscript.**

Example of reference: Stewart MK, Cookson BT. Non-genetic diversity shapes infectious capacity and host resistance. Trends Microbiol. 2012;20(10):461-466.

- Are there differences in bacterial growth between the isolates and that is why they have differences in virulence? Was this tested?

→ All seven isolates were tested in identical conditions for differences in bacterial growth through the OmniLog® system, which did not highlight any major difference between the strains over 72 hours. **We mentioned this in the revised manuscript.** The growth curves are displayed below **and were included as Supplementary Data in the revised manuscript.**

Only every 2 h data included

All data included

2.2. Pg 10, lns 242 – 254 . I was confused by the reasoning in this paragraph. The liver isolates and blood isolates share the same chromosomal characteristics, but the blood isolates were more exposed to the phage? Would this suggest that they were both exposed to the phage if similar phage resistance mutations were found in the liver isolates?

→ Blood and also liver *Pseudomonas aeruginosa* isolates are bound to have been exposed to phages. Phages were administered intra-lesionally (low volumes injected into the liver abscesses via pig-tail drainage catheter), daily, over a period of one week (lines 87-88). In addition, intravenously administered phages are likely to have reached the liver abscesses, as liver is one of the most commonly reported organs to accumulate phages delivered systemically. It filters foreign objects traveling in the blood, including phages. Upon systemic delivery, phages can reach the liver within minutes and can reach significant intra-hepatic titers.

In conclusion, our opinion is that blood isolates were most probably the most exposed to phages (through intravenous phage therapy), but liver isolates were definitely also exposed to phages, though possibly to a lesser extent (because intralesional phage therapy used very small volumes of phage solution and was probably not penetrating very far in the biliary tractus).

We modified a sentence in this part of the revised manuscript to stress the fact that phage therapy was mostly intravenous, hence the higher "phage exposition" of blood-circulating isolates.

Reference: Dąbrowska K. Phage therapy: What factors shape phage pharmacokinetics and bioavailability? Systematic and critical review. *Med Res Rev.* 2019;39(5):2000-2025.

2.3. I was confused by the reasoning pg 14 ln 328 – 330. Serum PIN was determined with ISP and not for PNM and 14-1. These are three different phages. Would this suggest this is phage-dependent?

→ Yes, susceptibility to develop PIN is seemingly both phage- and patient-dependent. Regarding phage-dependence, PIN susceptibility seems to be modulated, at least in part, by the intrinsic immunogenicity of each phage's specific surface epitopes, among which glycoproteins; as described in this reference for example: Majewska, J.; Beta, W.; Lecion, D.; Hodyra-Stefaniak, K.; Kłopot, A.; Ka'zmierzak, Z.; Miernikiewicz, P.; Piotrowicz, A.; Ciekot, J.; Owczarek, B.; et al. Oral application of T4 phage induces weak antibody production in the gut and in the blood. *Viruses* 2015, 7, 4783–4799.
We added a comment in that regard in the revised manuscript.

Such differences in intrinsic phage immunogenicity are, however, not sufficient to explain on their own the absence of PIN towards phages 14-1 and PNM in our case: indeed, at least one other unpublished case shows that intravenous administration of the same phage cocktail in an immunocompetent patient, which was also suffering from systemic *Pseudomonas aeruginosa* infection, can generate potent PIN against phages 14-1 and PNM in as little as two weeks of intravenous administration. These unpublished results are displayed below. Green fields indicate the two periods of time during which the patient was treated by intravenous phage therapy. Curves represent percentage of PIN over time for phages 14-1 and PNM. This highlights the importance of patient-dependent factors, in our case probably mostly immune suppression, in the susceptibility to develop PIN.

2.4. ISP eliciting an immune response in an immunosuppressed patient seems to be one of the more interesting aspects of this study. Could ISP have stimulated immunity and what contribution could that have made to the success phage therapy? Could you talk about this further in the text?

→ Phage-mediated immunomodulation can indeed have played a role in the progression of the child's clinical state under systemic phage therapy. The immunomodulatory properties of all three phages composing phage cocktail BFC1.10 have been previously described in the following reference : Van Belleghem, J. D., Clement, F., Merabishvili, M., Lavigne, R., & Vaneechoutte, M. (2017). Pro- and anti-inflammatory responses of peripheral blood mononuclear cells induced by Staphylococcus aureus and Pseudomonas aeruginosa phages. Scientific reports, 7(1), 8004. <https://doi.org/10.1038/s41598-017-08336-9>

These findings lead us to believe that the immunomodulatory properties of the administered phages are complex, but that their net effect is probably rather anti-inflammatory, and of comparable magnitude: phage ISP does not seem to display more marked immunomodulatory properties compared to phage 14-1 or phage PNM. So, phage ISP could indeed have interacted with the patient's immune system, potentially participating in a favorable evolution of his pro-inflammatory septic state, but this would be as likely for phage 14-1 or phage PNM in our opinion. In conclusion, phage ISP's ability to trigger PIN in this patient does not seem to be related to especially marked immunomodulatory properties, which phage 14-1 or phage PNM would not display. **We will mention this in the revised manuscript.**

Minor

2.5. Please include page numbers for the future.

→ We do not know where the problem might lay but all of us could see page numbers on the version that we downloaded via the editor, maybe it is a matter of PDF reader software version. We will make sure that page numbers are indeed included in the resubmitted manuscript, hoping they would show clearly this time. Please excuse us for the inconvenience should they still not appear.

2.6. Pg 2, Figure 1. This figure is difficult to read. Not sure if this is due to lower resolution. Please use higher resolution figures or larger, easier to read font (darker colors).

→ We increased the resolution of Figure 1.

2.7. Pg 3, lns 93-95 In the text it reads "...two transient P. aeruginosa translocations from liver abscesses to the bloodstream were observed on day 27 and day 30 after PT initiation..." however, this event is not noted on the timeline in Figure 1. It seems important, although transient, that an infection occurred during phage treatment. Please include the event on the timeline.

→ Figure 1 was modified to display these events.

2.8. Pg 12, ln 309. Should it say “data not shown” after this sentence? Is this data available?

→ The data is shown in Figure 7 as mentioned in the next sentence (line 310 in the original manuscript).

2.9. Pg 16, lns 405 and 417. Missing information for Manufacturer, city and country.

→ Missing information was added.

Reviewer #3 : response to comments

Dear Reviewer,

We thank you sincerely for your time and consideration in reviewing our manuscript. We took your remarks into consideration and believe this will strengthen our work. Please find a point-by-point response to your comments below.

Hoping this response brings you full satisfaction, we remain at your disposal for any further information.

Best regards,

Brieuc Van Nieuwenhuysse

3.1. The fitness costs of phage-resistance in bacteria from the liver abscess are ill-defined, perhaps because the assays used did not depend primarily on the trade-off observed.

→ Based on this comment we decided to carry out another type of assay to investigate the possible fitness costs – please see point 3.6.

3.2. The demonstrated lack of phage neutralisation might have ensured good delivery to site of action but phage kinetics were not directly measured. In fact, the dose given into the estimated blood volume is likely to yield a relatively low multiplicity of infection. The authors avoided this contentious area but it might have been mentioned briefly in Discussion – perhaps the demonstrated antibiotic synergy was particularly important in this case.

→ Indeed, we added a brief discussion to the revised conclusion.

Specific comments

3.3. Fig 1 is an excellent summary and overview of the clinical course and could be enlarged for easier cross-reference during the reading of the main text.

→ Fig. 1 was enlarged and the graphical resolution was increased.

3.4. Fig 2 is unnecessary

→ Fig. 2 was moved to Supplementary Data.

3.5. Re Fig 3(c) (Table) - the antimicrobial activity is presented as three separately determined and partially contradictory results for vegetative phase organisms and the genomic and genetic analysis implies an active mobile gene pool with cassette-based genes variably present in isolates from the same population at the same site at the same time. The data imply that the gene pool mobility is completely independent of and unaffected by the beta-lactam and aminoglycoside antibiotics used.

Lines 168-170 seem to imply that this gene traffic is relevant but perhaps the authors may wish to consider the comments below. Specifically - the integron-borne blaVIM (carbapenemase) gene cassettes, and the gene cassettes encoding aac 6' 29a and b which were first described with VIM-2 in *Pseudomonas*: (DOI: 10.1128/AAC.45.2.546-552.2001), all appear to be moving without regard to any selection pressure. The absence of the VIM-2 carbapenemase has no apparent effect on resistance to carbapenems, implying that the resistance is porin/ pump-mediated and, indeed, the finding of aztreonam susceptibility (the VIM carbapenemases being vulnerable to this synthetic monobactam) in all but isolates 2 and 6 implies that VIM-2 is not being selected at all. There is no annotated cephalosporinase such as PSE-2 identified in these but PDC-55, a class C type beta-lactamase normally found in *Pseudomonas*, is identified in all strains would be expected to hydrolyse aztreonam and might explain the carbapenem resistance phenotype in the presence of reduced permeability (eg an oprD lesion) or complementary efflux mechanism. Neither the ubiquitous (and in these isolates, persistent) aph 3'-1b (straA, also a very widespread class 1 integron gene cassette) nor the aac(6')-29a and 29b genes encode gentamicin resistance. The gene traffic described in Table /Fig 3(c) therefore just illustrate an active gene pool but not one that is convincingly influenced by any described selection pressure.

➔ We generally agree with the points made by the Reviewer, though we do not feel that the original manuscript makes contradictory claims. To make sure that we all share the same view on these points, here are our views:

- The original manuscript did not, in our opinion, claim that certain AMR genes had been specifically and dynamically selected by the application of certain antimicrobial therapeutics during the course of the case. However, we acknowledge that presenting the tables in Fig. 3 might be interpreted as such, while they should be viewed ultimately as purely descriptive. For this reason, **we will move previous tables "Fig. 3 b" and "Fig. 3 c" to Supplementary Data and will insert a comment in the revised manuscript** stating that we do not claim any correlation between the application of any antimicrobial and the selection of any AMR-related gene.

- The *Pseudomonas aeruginosa* clone infecting the patient is an ST111 (serotype O12), a well-known XDR clone of widespread geographical distribution. The anti-microbial resistance determinants in this high-risk clone are well documented and its expression of VIM-2 carbapenemase seems thoroughly reported. **We will mention this in the revised manuscript.** The presence (in most isolates) of an intermediate susceptibility to aztreonam is in our view a rather typical and reliable indicator that VIM-2 is indeed expressed in the isolate, and is likely responsible for the carbapenemase-resistant phenotype, but again, we do not want to convey the message that VIM-2 has been selected by any treatment during the case: it was most likely present and expressed from the beginning.

Example of recent reference : Del Barrio-Tofiño E, López-Causapé C, Oliver A. *Pseudomonas aeruginosa* epidemic high-risk clones and their association with horizontally-acquired β -lactamases: 2020 update. *Int J Antimicrob Agents* 56, 106196 (2020)

- Consistent with the previous reasoning that VIM-2 appears to be responsible for the carbapenem-resistance phenotype in most isolates, we double-checked our sequencing data and can confirm that all isolates of our ST111 clone are devoid of any *OprD* defective mutations that could represent an alternate genomic substrate to this carbapenem-resistant phenotype;

3.6. Lines 170-185. This is an interesting and important commentary – essentially, I think the authors are describing the simple trade-off of a type 4 pilus in a subpopulation within the liver abscess to avoid phage predation. We assume that the liver abscess is the primary site of infection and the finding of an ‘entire’ T4P expressing *P. aeruginosa* isolate in the blood at day 8 implies that this T4P tradeoff is a dominant subpopulation in the liver but not particularly successful as an invader. The known role of T4P in adhesion and biofilm initiation means that the choice of the galleria virulence model may not be ideal. Twitching motility assays might possibly be affected (although the implication from the genomic data is that the pilB rather than the pilT operon is impaired) but a growth competition assay or a virulence assay that does not depend on biofilm formation or pilin-mediated adhesion might be predicted to be inconclusive.

The discussion in lines 230-250 do not particularly address this and, even if an argument about non-specific fitness is to be made, a simple in vitro growth competition assay (between phage R and phage S) might be a better proof.

→ We agree with the Reviewer and performed two new assays to better evaluate the possibly associated fitness costs.

→ First, and upon conjoint request of another Reviewer, we checked for differences in the spontaneous growth curves of each isolate using the OmniLog system. No major difference in growth dynamics was observed. **This is mentioned in the revised manuscript.** These growth curves are displayed below and **were added to the revised manuscript as Supplementary Data.**

→ Second, and more relevant to your suggestion, we performed another virulence assay on HeLa cells to overcome some of the analytical limitations of the suboptimal *Galleria mellonella* model in this context. **The results of this new assay were added to the manuscript, as well as a dedicated Material & Methods section.** The absorbance at 570 nm was measured to evaluate the viability of HeLa cells. The highly virulent *P. aeruginosa* strain PA14 was used as a positive control, whereas HeLa cells without the addition of a *Pseudomonas* culture served as a negative control. A connected letter report was created to pairwise compare all isolates and the controls. Pa1BS was found to not significantly reduce the viability of the cells, compared to the negative control (see Figure below, which is now **part of the new Virulence assays Figure in the revised manuscript**). Similar to the *Galleria* assay results, no correlation was observed between the phage-resistant phenotype and the HeLa cell viability. Moreover, no correlation was observed between the data of the two virulence assays, with the exception of both assessments finding Pa1BS as the least virulent isolate of all. These new results do not support either the hypothesized correlation of phage resistance with reduced virulence. This could either mean that no significant phage-induced virulence tradeoffs occurred in this case, or that none of the experimental models were able to highlight them. The case-by-case choice of a pertinent experimental model to assess each case of putative phage-induced virulence tradeoffs (depending on bacterial species, phage infection mechanism and molecular target of the tradeoff) certainly appears as a challenge for future clinical research, and warrants further investigation.

Please note that we also chose to move previous Fig. 5 a-d to Supplementary to focus on results.

3.7. The phage-antibiotic synergy is nicely presented and are interesting, potentially important data. The convincing in vitro data imply that even a less effective phage and/or antibiotic might have impact when used in this way. Given the relatively unchanging phenotype over time and site for all three antibiotics, it might have been useful to show whether this synergy was at all evident in other than the initial isolate, as the reader would expect this not to be the case in the liver but presumably still true in the blood on day 8 of phage therapy.

→ We conducted the same OmniLog® PAS assay on the blood isolate from day 8 (Pa7_{BS}) and found indeed similar synergistic properties in the same growth conditions as the ones used for PAS assay on Pa1_{BS}, the only exception being a slightly less marked PAS with colistin. The curves are displayed below. **This comment was added to the revised manuscript and the new curves were added to the Phage-Antibiotic Synergy Figure with the original Pa1_{BS} results.** Please note that we also modified the labels in the original PAS Figure to correctly display the name of the isolate (Pa1_{BS} instead of S1).

3.8. Finally, the findings from phage neutralisation assays are important even if unsurprising. The relevance is presumably that the lack of neutralisation means that viral delivery from blood is guaranteed over time and that immune complex-mediated disease is avoided. No viral kinetics were measured so this finding in a critically ill child with a completely ablated immune system bears little on the case. The phage-antibiotic synergy may have been a saving grace in this clinical scenario but there are no drug or phage measurements in the blood available to directly show that. Nevertheless, this is an interesting and thought-provoking anecdote.

→ Agreeing with the Reviewer, we added a brief part addressing this in the revised manuscript, the same as the one motivated by their comment n°3.2.

REVIEWERS' COMMENTS

Reviewer #3 (Remarks to the Author):

The authors have responded to all the comments raised by this reviewer and have significantly added to the manuscript.

The assays used to assess bacterial fitness are still too insensitive (simple growth kinetics) and this remains a question mark but is now acknowledged in the text and is therefore acceptable.